# Estimates of the number and distribution of zero-dose and under-immunised children across remote-rural, urban, and conflict-affected settings in low and middle-income countries

Adelle Wigley[1]*, Josh Lorin[2], Dan Hogan[2], C. Edson Utazi[1], Brittany Hagedorn[3], Emily Dansereau[3], Andrew J. Tatem[1], Natalia Tejedor-Garavito[1]

**1** WorldPop, Geography and Environmental Science, University of Southampton, Highfield Campus, Southampton, United Kingdom, **2** Gavi, The Vaccine Alliance, Geneva, Switzerland, **3** Institute for Disease Modelling, Bill & Melinda Gates Foundation, Seattle, Washington, WA, United States of America

* a.s.wigley@soton.ac.uk

**Data Availability Statement:** The underlying data used in this work is publicly available via the

## Abstract

While there has been great success in increasing the coverage of new childhood vaccines globally, expanding routine immunization to reliably reach all children and communities has proven more challenging in many low- and middle-income countries. Achieving this requires vaccination strategies and interventions that identify and target those unvaccinated, guided by the most current and detailed data regarding their size and spatial distribution. Through the integration and harmonisation of a range of geospatial data sets, including population, vaccination coverage, travel-time, settlement type, and conflict locations. We estimated the numbers of children un- or under-vaccinated for measles and diphtheria-tetanus-pertussis, within remote-rural, urban, and conflict-affected locations. We explored how these numbers vary both nationally and sub-nationally, and assessed what proportions of children these categories captured, for 99 lower- and middle-income countries, for which data was available. We found that substantial heterogeneities exist both between and within countries. Of the total 14,030,486 children unvaccinated for DTP1, over 11% (1,656,757) of un- or under-vaccinated children were in remote-rural areas, more than 28% (2,849,671 and 1,129,915) in urban and peri-urban areas, and up to 60% in other settings, with nearly 40% found to be within 1-hour of the nearest town or city (though outside of urban/peri-urban areas). Of the total number of those unvaccinated, we estimated between 6% and 15% (826,976 to 2,068,785) to be in conflict-affected locations, based on either broad or narrow definitions of conflict. Our estimates provide insights into the inequalities in vaccination coverage, with the distributions of those unvaccinated varying significantly by country, region, and district. We demonstrate the need for further inquiry and characterisation of those unvaccinated, the thresholds used to define these, and for more country-specific and targeted approaches to defining such populations in the strategies and interventions used to reach them.

sources referenced in the Data and Methods section. The calculated estimates of un- and under-vaccinated children are available at https://www.doi.org/10.5258/SOTON/WP00728.

**Funding:** This work was supported by the Bill and Melinda Gates Foundation and Gavi, the Vaccine Alliance [Grant Number INV-002397 awarded to A.J.T, C.E.U and N.T.-G.] and in part by the Bill and Melinda Gates Foundation [Grant Number INV-007594 awarded to A.J.T, N.T.-G and A.W]. The funders had no role in study design, data collection and analysis, decision to publish, or preparation of the manuscript.

**Competing interests:** I have read the journal's policy and the authors of this manuscript have the following competing interests: DH and JW work for Gavi, the Vaccine Alliance, while BH and ED work for the Bill and Melinda Gates Foundation. The results and conclusions contained here are those of the authors and do not necessarily reflect the position or policies of Gavi, the Vaccine Alliance and the Bill and Melinda Gates Foundation. The authors declare no other competing interests.

## Introduction

While there has been great success in increasing the coverage of new childhood vaccines globally, expanding routine immunization services to reliably reach all children and communities has proven more challenging [1–4]. Certain populations remain un- or under-vaccinated in many LMICs [2, 5, 6] and infectious diseases still account for a large proportion of deaths among children [2, 7, 8]. Lack of accessibility to health care, poverty and overcrowding, as well as civil and political unrest, are all considered major contributors to the transmission of vaccine preventable diseases, and present significant challenges in reaching global coverage targets [9]. In 2019, 13.6 million children failed to receive even a first dose of the diphtheria-tetanus-pertussis vaccine (DTP1), rising significantly to 17.1 million in 2020, due to disruptions to health services amidst the ongoing coronavirus pandemic [10]. Moreover, the percentage of children receiving three doses of the diphtheria-tetanus-pertussis vaccine (DTP3) declined by 5 percent from 2019 to 2021, reflecting a sustained decline in DTP3 coverage [11].

Considering the substantial burden of vaccine-preventable diseases and the frequent occurrence of outbreaks among disadvantaged populations [2, 12] there have been continuous efforts in the health policy environment to design vaccination strategies and interventions that prioritize and target these population groups [3, 13, 14]. In 2017, senior leaders in the immunization sector formed the Equity Reference Group for Immunization (ERG), identifying remote-rural, conflict-affected and the urban poor as likely groups to be un- or under-vaccinated; also recognizing gender-related barriers as a cross-cutting issue [15, 16]. Gavi, the Vaccine Alliance, has adopted a new five-year strategy ("Gavi 5.0") for 2021–2025, with a vision of "Leaving no one behind with immunization" by increasing equitable and sustainable use of vaccines [17]. Further, the World Health Organization's Immunization Agenda 2030 sets a global vision and strategy for vaccines and immunization, committing to an ambitious target of reducing the number of zero-dose children by 50% by 2030 [18].

Despite these developments, estimates of how many children living in remote-rural, urban slum or conflict-affected settings, their vaccination rates and relative distributions within countries remain scarce, limiting robust resource allocation and strategy design [19]. Previous studies have revealed significant geographic inequalities in routine vaccination coverage across many LMICs, through analysis at the district level, helping to identify areas of low coverage and vaccine delivery system vulnerabilities [5, 8, 20–22]. However, to date, there has been no global scale analysis at these finer spatial resolutions focusing on the numbers of unvaccinated children in these key geographically marginalised populations [6, 9, 18]. Achieving equitable coverage, which has been identified as a core priority moving forward [17, 18, 23] needs to be guided by the most current and detailed evidence regarding the size, distribution, and characteristics of these key at risk groups [20, 21, 24–26], if current global immunization targets are to be met [17, 18].

Here, we present outputs from the integration of a range of geospatial datasets to estimate the number and distribution of zero-dose and under immunised children, across remote-rural, conflict-affected, and urban settings. We estimate the relative proportions and numbers of these key populations, to assess variabilities both between and within countries, to help guide global, national, and regional strategies and interventions, and to highlight those areas where children may remain unvaccinated. Given available data and to reflect key immunization program performance indicators, our study focuses on receipt of DTP1, DTP3, and the first dose of the measles containing vaccine (MCV1) among children under the age of one in 2019.

## Data and methods

We assembled geospatial datasets on the distributions of children under the age of one (U1) [27], vaccination coverage estimates of DTP1, DTP3 and MCV1 (VC) [5], estimated travel-time to the nearest major settlement [28], the location and timing of conflict-related fatalities [29], the distribution and extent of urban, peri-urban, and rural areas [30], and administrative boundaries at the second subnational level (ADM2) [31]. Note we use the term 'unvaccinated' throughout to refer to non-receipt of DTP1, DTP3 or MCV1.

Using vaccination coverage and population estimates, we first calculated both the total numbers of U1 and the numbers of unvaccinated U1. Then, using the travel-time, conflict, settlement type, and administrative boundary data, we estimated the numbers of unvaccinated U1 living in (i) remote-rural (and rural non-remote), (ii) conflict-affected (broad and narrow definitions, see below) and (iii) urban and peri-urban locations. The resulting datasets were summarised for 99 LMICs that had subnational boundaries at ADM2 for which data was available. The following outlines each of the steps in more detail.

Ethics approval for this article was obtained from the Ethics Committee at the Faculty of Environmental and Life Sciences, University of Southampton, United Kingdom (Ethics number: 50248). Consent to participate is not applicable.

**Mapping U1.** Estimates of the number of U1 at the 1x1 km grid square scale, were obtained from the WorldPop database for 2019 [27]. These were then summarised at ADM2 using the database of Global Administrative Areas (GADM) [31]. Detailed publications on the methods used to produce these estimates are available elsewhere [32–34]. Though in brief, machine learning methods in combination with a range of geospatial covariates, are used to disaggregate population counts from areal units to 1x1 km grid squares, and models based on census, survey and microdata used to further disaggregate these by age and sex.

**Mapping unvaccinated U1 for DTP1, DTP3 and MCV1.** To estimate the numbers of unvaccinated U1, vaccination coverage (VC) rates for DTP1, DTP3, and MCV1, were obtained from the Institute for Health Metrics and Evaluation (IHME) for 2019, at 5x5km resolution. Methods detailing the production of these datasets are described in Sbarra A. et al. [5]. The vaccination coverage datasets were re-sampled to 1x1km resolution, to align to the WorldPop estimates of U1 [27], and the numbers of unvaccinated U1 per 1x1 km grid square subsequently calculated as: (1-VC) X U1, before summarising by ADM2 using the GADM administrative boundaries.

**Mapping unvaccinated U1 in urban and peri-urban areas.** To estimate the numbers of unvaccinated U1 in urban and peri-urban areas, we used the Global Human Settlement Layer (GHS-SMOD), which delineates and classifies settlement typologies, using both population size and built-up area densities [30]. To define urban areas, we used the *urban centre* grid cell category, and to define peri-urban we used the *dense urban*, *semi-dense urban*, *and sub-urban* grid cell categories, assigning each a grid square value of 1, with these categories making up the 'urban domain' as specified in the GHS-SMOD. The data were harmonized and resampled to the same resolution as the WorldPop data, before multiplying with the population and vaccination coverage data, to estimate the numbers of unvaccinated U1 in urban and peri-urban areas at the grid square level. The data were then summarised at ADM2, using the GADM administrative boundaries.

Whilst identified as a key population group likely to be un- or under-vaccinated, the availability and reliability of a global data set on urban slum locations makes it challenging to estimate the proportion of children not receiving vaccinations in urban slums. This is further complicated by the spatial resolution of the IHME immunisation coverage estimates, produced at the 5x5 km2 scale, which is larger than many areas of urban slums, thus having the potential to overestimate the numbers of unvaccinated children living in these areas. For these reasons

we have not included estimates of the numbers of unvaccinated U1 living in urban slums in this analysis.

**Mapping unvaccinated U1 in remote-rural areas.**   To estimate the numbers of unvaccinated U1 in remote-rural areas, travel-time estimates to the nearest town/city of 50,000 people or more [28] were used to define *remote* areas, and the GHS-SMOD used to define *rural* areas [30]. We classified as *remote* those areas where the travel-time was 180 minutes or more to the nearest town/city [28, 35, 36] and as *rural*, those areas in the GHS-SMOD with a rural grid square classification, using only grid cells that were classified as both *rural* and *remote*, then assigning these a grid square value of 1. The data were harmonized and resampled to match the WorldPop data, before multiplying with the population and vaccination coverage data, to estimate the numbers of unvaccinated U1 in remote-rural areas at the grid square level. The data were then summarised at ADM2, using the GADM administrative boundaries.

**Mapping unvaccinated U1 in rural (non-remote) areas.**   The remainder of U1 (those not classified as urban, peri-urban, or remote-rural areas) were considered rural (non-remote). As in Weiss et al. (2020) [45], we use travel-time categories to summarise the numbers of unvaccinated U1 in rural (non-remote) areas, defined as rural areas within 180 minutes of the nearest town or city, classifying the travel-times into 0 to 60, 60 to 120, and 120 to 180 minute categories, and assigning each a grid square value of one. The data were harmonized and resampled to match the WorldPop data, before multiplying with the population data and vaccination coverage data, to estimate the numbers of unvaccinated U1 in rural (non-remote) areas at the grid square level. The data were then summarised at ADM2, using the GADM administrative boundaries.

**Mapping unvaccinated U1 in conflict-affected areas.**   To estimate the numbers of unvaccinated U1 in conflict-affected areas, geo-located conflict locations were obtained from the Armed Conflict Location & Event Data Project (ACLED) [29], for conflicts resulting in violent deaths (battles, explosions/remote violence, and violence against citizens), two years prior to 2019. From these we selected conflicts resulting in at least one fatality and used these to select ADM2 units experiencing at least one conflict, to define conflict-affected areas. The data were then summarised using only the conflict affected ADM2 units. After evaluation and comparisons to the perceived conflict situation, likely impacts on service delivery, and with reference to current literature on defining exposure to conflict, this definition was refined further. Using thresholds of at least 30 and 300 fatalities per million people to classify districts as being conflicted-affected according to both broad and narrow definitions of conflict, respectively.

## Results

We found substantial heterogeneities in the spatial distribution and setting (or context) of unvaccinated U1, both among and within countries, with a large proportion of them found to be concentrated within a small number of countries. Whilst many of those unvaccinated were found to be in remote-rural (11.8%), conflict-affected (5.9% to 14.7%), and urban locations (20.3%), a large proportion were found in settings other than these (59.7%), with nearly 40% found to be in rural (non-remote) locations within 1-hour of the nearest town or city (not including urban and peri-urban areas) (DTP1).

### Geographical distributions of unvaccinated U1

We first examined the geographical distributions of unvaccinated U1. Fig 1 shows the estimated proportion of U1 who did not receive the first dose of the DTP vaccine in 2019 at ADM2 and highlights the substantial differences in vaccination coverage both among and within countries. S1 and S2 Figs, show the same for DTP3 and MCV1 and display broadly similar spatial patterns. The largest numbers of unvaccinated U1 for all three doses were in India

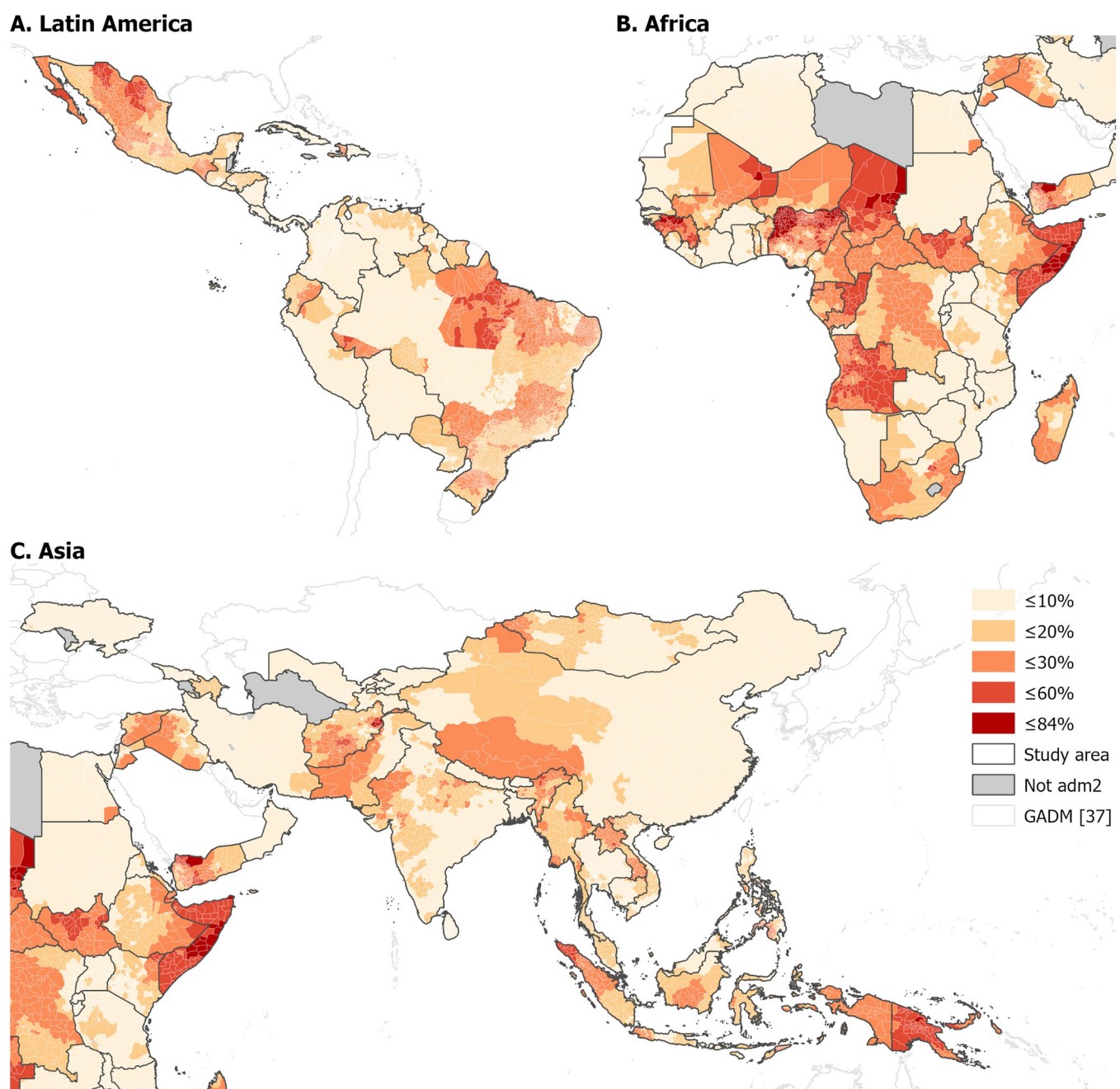

**A. Latin America**

**B. Africa**

**C. Asia**

≤10%
≤20%
≤30%
≤60%
≤84%
Study area
Not adm2
GADM [37]

**Fig 1.** Proportion of children under 1 year of age estimated to have not received the first dose of the DTP vaccine in 2019 at GADM administrative level 2 [31, 37] for Latin America (A), Africa (B), and Asia (C).

and Nigeria, whereas the largest proportions were in Somalia, Papua New Guinea, Guinea, Chad, and Angola (S3 Fig).

## Breakdown of unvaccinated U1 by setting

Next, we examined the numbers and proportions of unvaccinated U1 residing in remote-rural, conflict-affected (broad and narrow), urban and peri-urban, and rural (non-remote) locations at the global, national, and subnational levels to further characterise those unvaccinated.

**Table 1. Global distribution of unvaccinated children under 1 year of age (percentage and number) in 2019 broken down by geographical setting, vaccine dose, and conflict for all countries in the analysis (Note: Conflict can overlap with other settings in the table).**

| Geography | DTP1 | DTP3 | MCV1 | U1 |
|---|---|---|---|---|
| | N (%) | N (%) | N (%) | N (%) |
| | (N = 14,030,486) | (N = 23,275,803) | (N = 20,237,173) | (N = 119,142,068) |
| **Urban** | 20.3 (2,849,671) | 19.8 (4,617,866) | 19.2 (3,879,778) | 26.6 (31,655,602) |
| **Peri-urban** | 8.1 (1,129,915) | 7.4 (1,719,382) | 7.1 (1,438,851) | 11.5 (13,723,037) |
| **Rural (non-remote)** | 59.7 (8,373,582) | 59.5 (13,848,978) | 60.7 (12,286,212) | 55.0 (65,470,911) |
| **< 1 hr** | 41.8 (5,868,071) | 40.3 (9,369,570) | 41.2 (8,336,881) | 40.9 (48,785,917) |
| **1 to 2 hrs** | 12.1 (1,694,761) | 12.8 (2,969,143) | 13.0 (2,626,086) | 9.7 (11,597,762) |
| **2 to 3 hrs** | 5.8 (810,750) | 6.5 (1,510,265) | 6.5 (1,323,245) | 4.3 (5,087,232) |
| **Rural (remote) > 3 hrs** | 11.8 (1,656,757) | 13.1 (3,059,570) | 12.9 (2,603,406) | 6.8 (8,054,098) |
| **Conflict (narrow to broad)** | 5.9 to 14.7 (826,976 to 2,068,785) | 5.9 to 16.5 (1,362,520 to 3,851,618) | 6.3 to 18.1 (1,270,709 to 3,658,501) | 2.6 to 9.1 (3,075,760 to 10,818,029) |

Table 1 shows the distribution of unvaccinated U1 broken down by geographical setting, vaccine dose, and conflict for all countries in the analysis. We found that more than 19% lived in urban areas, with a smaller proportion in peri-urban areas, whilst more than 11% lived in remote-rural areas more than 3 hours travel-time from the nearest town or city. Around 60% of those unvaccinated lived in other rural areas, not classed as remote, urban, or peri-urban (rural non-remote). Table 1 also shows the percentage of unvaccinated U1 considered to be conflict-affected, which was estimated to be between 6% and 18%, depending on the definition of conflict used (narrow and broad).

## Unvaccinated U1 in remote-rural locations

Fig 2 shows where the unvaccinated U1 in remote-rural areas were concentrated, by mapping the proportions of U1 unvaccinated for DTP1 (of the total unvaccinated) at ADM2. The map highlights countries where significant proportions of U1 were estimated to be unvaccinated, including the Central African Republic, Republic of Congo, and Mauritania, as identified above. However, it additionally reveals regions not previously identified at the national level, for example large areas of northern Chad, central DRC, and South-eastern Angola/Eastern Botswana. The distributions of unvaccinated U1 in remote-rural areas were similar for all three vaccine indicators, though generally appeared slightly lower for MCV1 (S4 and S5 Figs).

S6–S8 Figs, show the top 20 ranked countries, by the estimated proportion (of the total unvaccinated) and corresponding total number of U1 not receiving DTP1/DTP3/MCV1 in remote-rural areas. In terms of proportion, Madagascar, Mauritania, Central African Republic, Papua New Guinea, and Republic of Congo were in the top 5 ranked countries for both DTP1 and DTP3, where around 60% to 70% of unvaccinated U1 were estimated to be in remote-rural locations. These figures are similar for MCV1, where around 60 to 66% of U1 unvaccinated for MCV1 were found to be in remote-rural locations, though Eritrea additionally ranks in the top five countries, rather than the Republic of Congo. However, when considering absolute numbers DR Congo was estimated to have the greatest number of unvaccinated U1 in remote-rural locations for all vaccine doses (on the order of 202,025 children, for DTP1), though Chad and Madagascar remain high in the rankings.

## Unvaccinated U1 in urban and peri-urban locations

Of the total number of U1 around 20% were found to be in urban areas. S9–S11 Figs, show where the unvaccinated U1 in urban locations were concentrated, by mapping the proportions

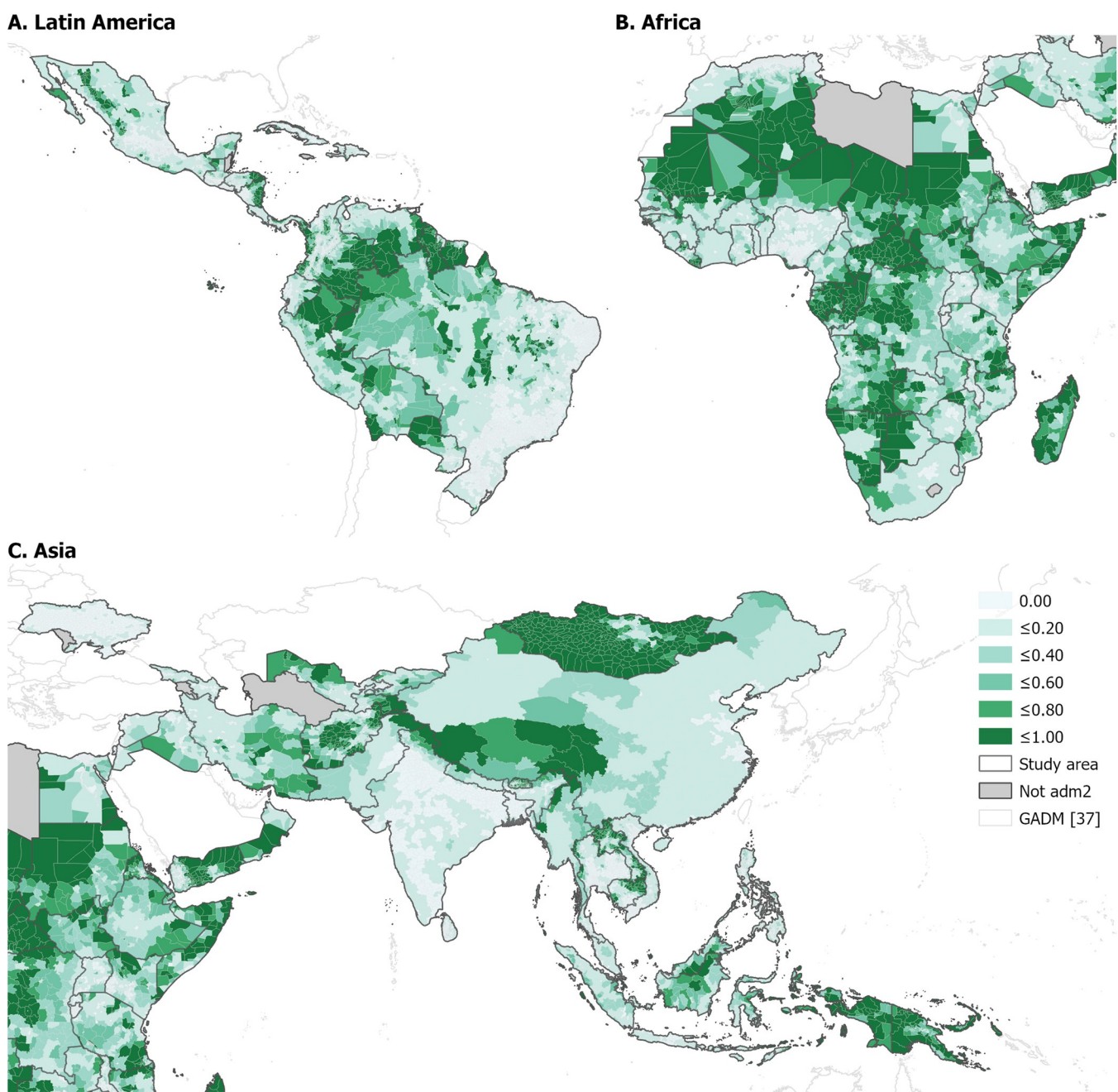

**Fig 2.** Proportion of children under 1 year of age in remote-rural areas estimated to have not received the first dose of the DTP vaccine in 2019 at GADM administrative level 2 [31, 37] for Latin America (A), Africa (B), and Asia (C).

of unvaccinated U1 (of the total unvaccinated) at ADM2. The map highlights key regions such as north-west Mexico, and central Brazil, which show large heterogeneity in the distributions of those unvaccinated. However, in countries with larger urban populations, such as India and China, higher proportions of unvaccinated U1 were observed in urban areas across a greater number of administrative units compared to countries with lower urban populations. These distributions are similar for all three vaccine doses.

S12–S14 Figs, show the top 20 countries by estimated proportions and numbers of U1 not receiving DTP1/DTP3/MCV1 in urban areas. In terms of the proportions of unvaccinated U1, Egypt and Mexico had the largest, with nearly 60% of U1 not receiving DTP1 in urban areas, and when considering absolute numbers Mexico also had the largest numbers of unvaccinated U1 in urban areas, with Brazil also ranking high.

S15–S17 Figs, map the percentage distribution of U1 not receiving DTP1/DTP3/MCV1 (of the total unvaccinated) within peri-urban areas at ADM2. The maps highlight key regions where the unvaccinated U1 were located, though with a different pattern in the distribution of those unvaccinated compared to the distributions of unvaccinated U1 in urban areas, with larger proportions observed across many areas throughout Brazil, South Africa, and China in particular. These distributions are similar for all three doses examined.

## Unvaccinated U1 in rural (non-remote) locations

With a large proportion of unvaccinated U1 estimated to be in rural areas not classed as remote, urban or peri-urban (rural non-remote), this category was broken down further by travel-time to the nearest town or city, into 0 to 60, 60 to 120, and 120 to 180 minute zones, to see how the numbers of unvaccinated U1 were distributed within these areas.

S18 to S23 Figs, map the percentage breakdown of U1 not receiving DTP1/DTP3/MCV1 within rural (non-remote) areas at the second administrative level, within 0 to 60 minutes, 60 to 120, and 120 to 180 mins of the nearest town or city, in Pakistan, India, Nigeria, and Ethiopia. Although some spatial variation can be observed between countries, a large proportion of unvaccinated U1 within rural (non-remote) areas were found to be located within 0 to 60 minutes of the nearest town or city. Most notably within India and Nigeria, where many districts are found to have the majority of those unvaccinated living in rural (non-remote) areas, within 1 hour of the nearest town or city.

## Unvaccinated U1 in conflict-affected locations

Fig 3 shows where the unvaccinated U1 (DTP1) in conflict-affected locations were concentrated according to the broad definition of conflict (at least 30 fatalities per million people) at ADM2 (S24 and S25 Figs). The map highlights conflict-affected regions where large proportions of U1 were unvaccinated, including much of Somalia (52%), Chad (46%), northeast Nigeria (34%), large parts of Central African Republic (26%), and Afghanistan (17%).

S26–S28 Figs, show the top 20 countries, by estimated proportion and numbers of U1 not receiving DTP1/DTP3/MCV1 in conflict-affected areas according to the broad definition. In terms of the proportions of U1 unvaccinated, Syria, South Sudan and Afghanistan had the highest prevalence of unvaccinated U1, with over 80% of U1 not receiving DTP1 found to be in conflict-affected areas. However, when considering absolute numbers, Nigeria had the largest numbers of unvaccinated U1 in conflict -affected areas, though Somalia, South Sudan, and DRC also ranked high. Note that conflict-affected unvaccinated U1 are disbursed across all other geographic settings (urban, peri-urban, remote-rural, and rural non-remote) and included all those unvaccinated within a conflict affected ADM2 unit.

## Unvaccinated U1 across all settings

Fig 4 shows the proportion of U1 not receiving DTP1, broken down by country and urban/rural setting. It highlights the variations that exist between countries, with some countries (e.g. Egypt, South Africa) found to have the largest proportion of U1 not receiving DTP1 in urban areas, and other countries (e.g. Mauritania, Madagascar) found to have substantial numbers in

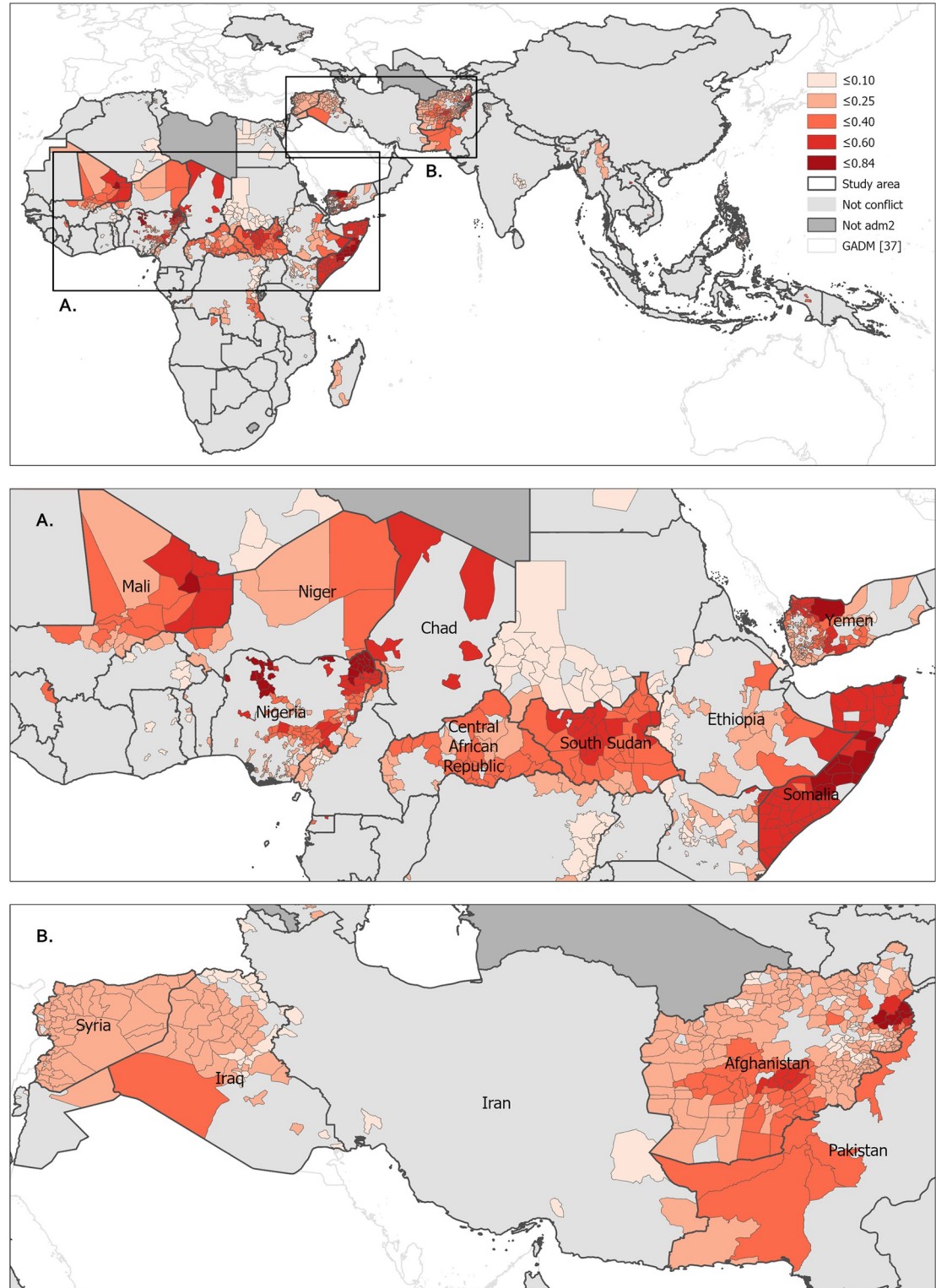

**Fig 3.** Proportion of children under 1 years of age in conflict-affected areas (broad definition) estimated to have not received the first dose of the DTP vaccine in 2019 at GADM administrative level 2 [31, 37] for Africa and Asia, with close ups shown for northern sub-Saharan Africa (A), and the Middle East/West Asia region (B).

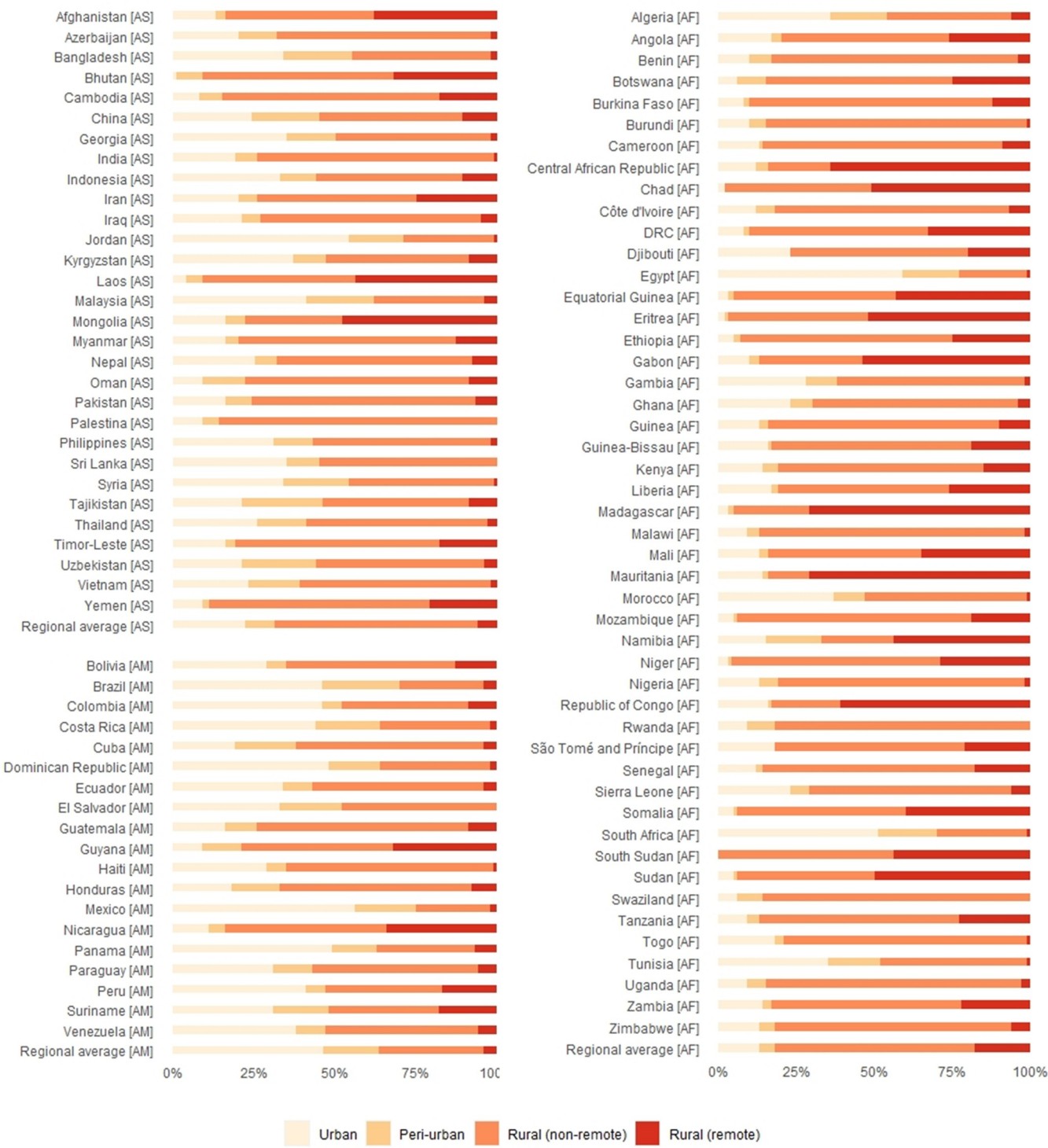

**Fig 4. Plot showing the estimated breakdown of children under 1 year of age not receiving DTP1 in 2019 by urban/peri-urban/rural (non-remote)/rural (remote) characteristics, for all countries in the study area in Latin America (AM), Africa (AF), and Asia (AS), and also regional averages.**

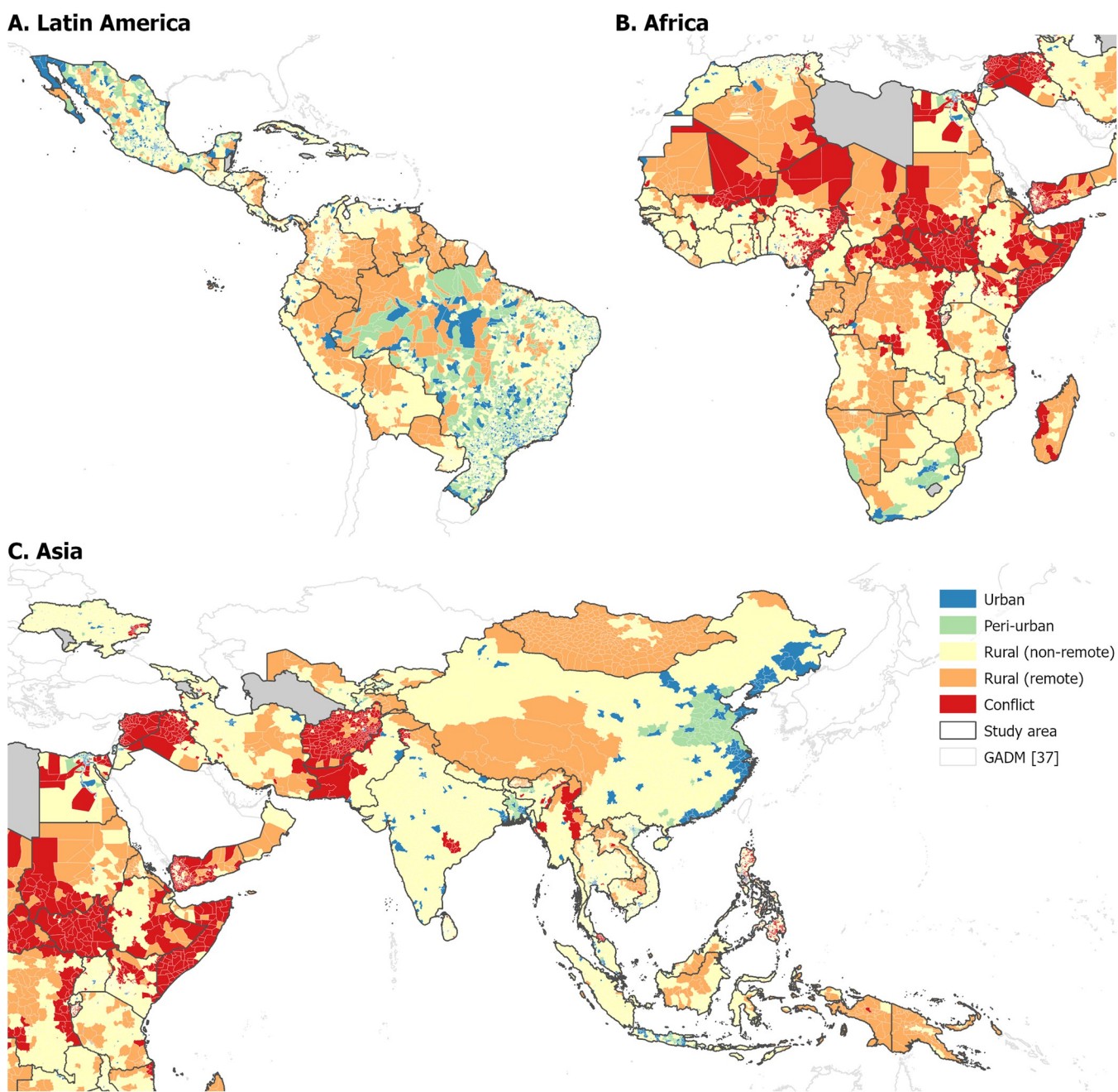

**Fig 5.** Map highlighting the geographical setting with the estimated greatest number of children under 1 years of age in 2019 not receiving DTP1 at GADM administrative level 2 [31, 37] for Latin America (A), Africa (B), and Asia (C).

remote-rural areas. Similar proportions for urban/rural settings were observed for the different vaccines (S29 and S30 Figs).

Fig 5 maps the setting where the greatest number of unvaccinated U1 (DTP1) were found, at ADM2, for urban, peri-urban, rural (non-remote), rural (remote), and conflict areas. Again, large heterogeneities were observed both among and within countries, with different characteristics of those unvaccinated observed at both national and sub-national levels. For example, in India, the majority of unvaccinated U1 were found to be in rural (non-remote) areas.

However, within Africa, most of those unvaccinated were found to be in conflict-affected, rural (remote) locations, and rural (non-remote) locations. In Latin America, we observed significant proportions of unvaccinated U1 in rural (non-remote) and rural (remote) areas, however in Brazil, higher numbers were estimated to be found in urban and peri-urban areas (S31 and S32 Figs).

## Discussion

If global targets to reduce the number of zero-dose children are to be met [18], then immunisation programmes need to identify and reach those children [6, 17, 18, 23, 24, 38, 39]. This requires the use of data driven actionable evidence, regarding the location and characteristics of those unvaccinated, and the design of tailored vaccination strategies appropriate for a given context [9, 18, 20, 21, 24, 25]. This study integrates a range of geospatial datasets [5, 27, 29, 30] to estimate the numbers of unvaccinated children living within key at-risk settings, including urban, remote-rural, and conflict affected locations, at the sub-national level. It is complementary to country-specific work, which identifies community and other household and individual level factors that characterise zero-dose populations and quantifies their relationship to vaccination coverage [26].

We find that coverage varies widely among and within countries, with substantial heterogeneities existing in the context and spatial distributions of those unvaccinated. Considering the settings identified by the ERG, we estimate that over 11% of zero-dose children are living in remote-rural areas, more than 26% in urban and peri-urban areas, and between 6 and 18% in conflict-affected locations. Altogether, this means that nearly 40% of zero-dose children are living in remote-rural and urban settings, and nearly 20% in conflict settings. Yet, whilst many of those unvaccinated are found to be living in these key at-risk settings, a significant proportion are located within areas other than these, in areas with lower population and/or built-up density than an urban or peri-urban setting. These findings highlight the variability of immunization coverage, both geographically and in terms of the characteristics of those populations, demonstrating the need for both country and regional specific strategies for immunization, that consider country size, available resources, and the context in which people live [17, 18], as well as further identification and characterisation of those most at risk of missing out on immunisation [17, 18].

There is a need for broader ranging yet tailored interventions, particularly while there are ongoing immunisation system vulnerabilities due to the COVID-19 pandemic [24]. One of the key principles guiding Universal Health Coverage is community designed and driven implementation of services, and there has been previous success in strategies that prioritise community engagement to identify and reach vulnerable populations [9, 40], as well improved public-private partnerships and microplanning to help identify and reach vulnerable populations [9, 40]. Additionally, the ERG recommends a range of targeted data-driven interventions that can help to address vulnerable populations in difficult to identify, hard to reach, and dangerous and fragile settings [40]. Which emphasise the need for timely and actionable data on vulnerable populations, through the use and implementation of electronic immunisation registries, dashboards, and visualisations, that allow current data to be collected and aggregated to the appropriate level for informative decision making [41–43].

Whilst we aim to provide the most detailed and up to date analysis of un- and under- vaccinated children characterised as most at risk, the analysis is limited by the choice and definition of those settings, data quality limitations, and both the availability of and ability to link together data from a variety of disparate sources. Regarding the choice of the settings, we chose to focus on key at risk populations, as identified by the ERG [40] to begin to characterise

those most at risk from un- and under-vaccination. However, as highlighted by the results, nearly 60% are found to be living outside of these settings. There is, therefore, scope to expand on and further characterise the key geographical settings in which the most disadvantaged populations are located, as well as the consideration of non-geographic dimensions, such as gender related barriers. Further work could look to examine the numbers of zero-dose children living outside of urban, remote-rural, and conflict-affected settings, to explore other potentially relevant variables that may help to explain barriers to immunisation [18, 44]. For example, when measuring remoteness, using travel-time to the nearest health facility [45], as opposed to the nearest town or city, which may be a more relevant measure in assessing the effect of geographical accessibility on vaccination coverage [46, 47].

Additionally, defining what is urban or rural can be challenging, with governments typically establishing their own definitions, making it difficult to meaningfully compare indicators across national borders [48]. For this study, we use the most current and recent globally available standard definitions of urban and rural [30]. However, the datasets feeding into these decisions are typically continuous in nature and so the choice of threshold or data to use naturally affects the estimates produced [49], and such simple classifications may unduly lead to the concealment of smaller areas of geographical inequality [50]. There is additionally the challenge of defining informal urban settlements, without access to globally available comparable data. An attempt was made to use accessibility to utility services [51] as a proxy for informal settlements, though comparison to alternative slum data proved validation difficult.

Furthermore, there is scope for exploring the sensitivity of these definitions, such as evaluating the adjustment of threshold values and the comparison of alternative data sets, to examine how much of an effect each has on the estimated numbers of unvaccinated children found to be living within these settings [52]. More recently, there have been collaborative efforts to address these issues, such as the development of a new global definition of urban and rural areas, using the 'Degree of Urbanisation,' to facilitate international comparisons. This method considers the urban-rural continuum, using three classes of 'cities,' 'towns & semi-dense areas,' and 'rural areas,' instead of two, as well as population size and density, and considers the functional urban area around each city [53]. Such data may provide more insight into the numbers of zero-dose children found to be living outside urban areas, though within 1 hour to the nearest town or city.

The analysis links together data from a variety of disparate sources, each with their own limitations of use that need to be considered. Whilst we aim to assemble to most recent and spatially detailed data available, certain compromises need to be made to facilitate comparability between the data sets and global coverage. For example, whilst more recent and higher resolution population estimates are available elsewhere [27], the vaccination coverage estimates are available at 5km, and it was deemed sufficient to conduct the analysis using the 1km population estimates within the same period of the vaccination coverage estimates. Additionally, whilst the accuracy and quality of the vaccination coverage estimates have been assessed elsewhere [5], uncertainties remain as to whether survey samples have been conducted in areas of conflict, for instance. Additionally, while we consider remoteness as travel-time to the nearest town/or city [28], one could also consider travel-time to the nearest health facility or hospital, or distance to major roads, which may be of interest in further analysis. Another key factor influencing the estimated outputs is the choice of administrative boundaries used to summarise the numbers of unvaccinated children, as naturally, the use of different or updated boundaries inherently affects how these numbers are aggregated, as demonstrated by the Modifiable Areal Unit Problem (MAUP), which describes how spatial summary measures are inherently influenced by the boundaries that they are reported at [54]. Finally, whilst we attempt to capture conflict using both broad and narrow definitions, the data available [29] largely relies on

media coverage, and whilst we aim to consider two years prior to the vaccination coverage estimates, areas of conflict or insecurity can change significantly over time.

Despite the inherent uncertainties observed in the data quality, definitions and thresholds applied, we present here the first available sub-national estimates, across 99 low- and middle-income countries, on the distributions and numbers of children not receiving vaccine doses in key at-risk settings. These baseline estimates provide significant insights into the inequalities in vaccination coverage, that can assist policy makers in the development of effective immunisation strategies that identify and reach all zero-dose children and missed communities [17]. We demonstrate additionally the need for further inquiry and characterisation of those un- and under immunised, the thresholds used to define these, and for more country-specific and targeted approaches to immunisation.

## Supporting information

**S1 Fig.** Proportion of children under 1 year of age estimated to have not received the third dose of the DTP vaccine in 2019 at administrative level 2 [31, 37] for Latin America (A), Africa (B), and Asia (C).
(TIF)

**S2 Fig.** Proportion of children under 1 year of age estimated to have not received the first dose of the MCV vaccine in 2019 at administrative level 2 [31, 37] for Latin America (A), Africa (B), and Asia (C).
(TIF)

**S3 Fig.** Graphs showing the top ten countries for children under 1 year of age not receiving DTP1/DTP3/MCV1 in 2019 measured by (left hand side) total number and (right hand side) proportion of the total children under 1 year of age; broken down by (a) first dose of DTP vaccine, (b) third dose of DTP vaccine; (c) first dose of MCV.
(TIF)

**S4 Fig.** Proportion of children under 1 year of age in remote-rural areas estimated to have not received the third dose of the DTP vaccine in 2019 at administrative level 2 [31, 37] for Latin America (A), Africa (B), and Asia (C).
(TIF)

**S5 Fig.** Proportion of children under 1 year of age in remote-rural areas estimated to have not received the first dose of the MCV vaccine in 2019 at administrative level 2 [31, 37] for Latin America (A), Africa (B), and Asia (C).
(TIF)

**S6 Fig. Proportion and total number of children under 1 year of age in remote-rural areas estimated to have not received the first dose of the DTP vaccine in 2019 for the top 20 highest ranking countries.**
(TIF)

**S7 Fig. Proportion and total number of children under 1 year of age in remote-rural areas estimated to have not received the third dose of the DTP vaccine in 2019 for the top 20 highest ranking countries.**
(TIF)

**S8 Fig. Proportion and total number of children under 1 year of age in remote-rural areas estimated to have not received the first dose of the MCV vaccine in 2019 for the top 20**

**highest ranking countries.**
(TIF)

**S9 Fig.** Proportion of children under 1 year of age in urban areas estimated to have not received the first dose of the DTP vaccine in 2019 at administrative level 2 [31, 37] for Latin America (A), Africa (B), and Asia (C).
(TIF)

**S10 Fig.** Proportion of children under 1 year of age in urban areas estimated to have not received the third dose of the DTP vaccine in 2019 at administrative level 2 [31, 37] for Latin America (A), Africa (B), and Asia (C).
(TIF)

**S11 Fig.** Proportion of children under 1 year of age in remote-rural areas estimated to have not received the first dose of the MCV vaccine in 2019 at administrative level 2 [31, 37] for Latin America (A), Africa (B), and Asia (C).
(TIF)

**S12 Fig. Proportion and total number of children under 1 year of age in urban areas estimated to have not received the first dose of the DTP vaccine in 2019 for the top 20 highest ranking countries.**
(TIF)

**S13 Fig. Proportion and total number of children under 1 year of age in urban areas estimated to have not received the third dose of the DTP vaccine in 2019 for the top 20 highest ranking countries.**
(TIF)

**S14 Fig. Proportion and total number of children under 1 year of age in urban areas estimated to have not received the first dose of the MCV vaccine in 2019 for the top 20 highest ranking countries.**
(TIF)

**S15 Fig.** Proportion of children under 1 year of age in peri-urban areas estimated to have not received the first dose of the DTP vaccine in 2019 at administrative level 2 [31, 37] for Latin America (A), Africa (B), and Asia (C).
(TIF)

**S16 Fig.** Proportion of children under 1 year of age in peri-urban areas estimated to have not received the third dose of the DTP vaccine in 2019 at administrative level 2 [31, 37] for Latin America (A), Africa (B), and Asia (C).
(TIF)

**S17 Fig.** Proportion of children under 1 year of age in peri-urban areas estimated to have not received the first dose of the MCV vaccine in 2019 at administrative level 2 [31, 37] for Latin America (A), Africa (B), and Asia (C).
(TIF)

**S18 Fig.** Proportion of children under 1 year of age unvaccinated for DTP1 in rural non-remote areas (Pakistan and India [31, 37]), within 0 to 60 (A), 60 to 120 (B), and 120 to 180 (C) minutes of the nearest town or city of 500,000 people or more.
(TIF)

**S19 Fig.** Proportion of children under 1 year of age unvaccinated for DTP1 in rural non-remote areas (Nigeria and Ethiopia [31, 37]), within 0 to 60 (A), 60 to 120 (B), and 120 to 180 (C) minutes of the nearest town or city of 500,000 people or more.
(TIF)

**S20 Fig.** Proportion of children under 1 year of age unvaccinated for DTP3 in rural non-remote areas (Pakistan and India [31, 37]), within 0 to 60 (A), 60 to 120 (B), and 120 to 180 (C) minutes of the nearest town or city of 500,000 people or more.
(TIF)

**S21 Fig.** Proportion of children under 1 year of age unvaccinated for DTP3 in rural non-remote areas (Nigeria and Ethiopia [31, 37]), within 0 to 60 (A), 60 to 120 (B), and 120 to 180 (C) minutes of the nearest town or city of 500,000 people or more.
(TIF)

**S22 Fig.** Proportion of children under 1 year of age unvaccinated for MCV1 in rural non-remote areas (Pakistan and India [31, 37]), within 0 to 60 (A), 60 to 120 (B), and 120 to 180 (C) minutes of the nearest town or city of 500,000 people or more.
(TIF)

**S23 Fig.** Proportion of children under 1 year of age unvaccinated for MCV1 in rural non-remote areas (Nigeria and Ethiopia [31, 37]), within 0 to 60 (A), 60 to 120 (B), and 120 to 180 (C) minutes of the nearest town or city of 500,000 people or more.
(TIF)

**S24 Fig.** Proportion of children under 1 year of age in conflict-affected areas (broad definition) estimated to have not received the third dose of the DTP vaccine in 2019 at administrative level 2 for Africa and Asia [31, 37], with close ups shown for northern sub-Saharan Africa (A), and the Middle East/West Asia region (B).
(TIF)

**S25 Fig.** Proportion of children under 1 year of age in conflict-affected areas (broad definition) estimated to have not received the first dose of the MCV vaccine in 2019 at administrative level 2 for Africa and Asia [31, 37], with close ups shown for northern sub-Saharan Africa (A), and the Middle East/West Asia region (B).
(TIF)

**S26 Fig. Proportion and total number of children under 1 year of age in conflict-affected areas (broad definition) estimated to have not received the first dose of the DTP vaccine in 2019 for the top 20 highest ranking countries.**
(TIF)

**S27 Fig. Proportion and total number of children under 1 year of age in conflict-affected areas (broad definition) estimated to have not received the third dose of the DTP vaccine in 2019 for the top 20 highest ranking countries.**
(TIF)

**S28 Fig. Proportion and total number of children under 1 year of age in conflict-affected areas (broad definition) estimated to have not received the first dose of the MCV vaccine in 2019 for the top 20 highest ranking countries.**
(TIF)

**S29 Fig. Plot showing the estimated breakdown of children under 1 year of age not receiving DTP3 in 2019 by urban/peri-urban/rural (non-remote)/rural (remote) characteristics,**

for all countries in the study area in Latin America (AM), Africa (AF), and Asia (AS), and also regional averages.
(TIF)

**S30 Fig. Plot showing the estimated breakdown of children under 1 year of age not receiving MCV1 in 2019 by urban/peri-urban/rural (non-remote)/rural (remote) characteristics, for all countries in the study area in Latin America (AM), Africa (AF), and Asia (AS), and also regional averages.**
(TIF)

**S31 Fig.** Map highlighting the geographical setting with the estimated greatest number of children under 1 years of age in 2019 not receiving DTP3 at administrative level 2 [31, 37] for Latin America (A), Africa (B), and Asia (C).
(TIF)

**S32 Fig.** Map highlighting the geographical setting with the estimated greatest number of children under 1 years of age in 2019 not receiving MCV1 at administrative level 2 [31, 37] for Latin America (A), Africa (B), and Asia (C).
(TIF)

## Acknowledgments

We would like to acknowledge the support, data provision and insights provided by John Mosser at the Institute for Health Metrics and Evaluation (IHME), University of Washington, and Francesco Checchi at the London School of Hygiene & Tropical Medicine (LSHTM).

## Author Contributions

**Conceptualization:** Adelle Wigley, Josh Lorin, Dan Hogan, C. Edson Utazi, Brittany Hagedorn, Emily Dansereau, Andrew J. Tatem, Natalia Tejedor-Garavito.

**Data curation:** Adelle Wigley, Josh Lorin, Natalia Tejedor-Garavito.

**Formal analysis:** Adelle Wigley, Josh Lorin, Natalia Tejedor-Garavito.

**Funding acquisition:** C. Edson Utazi, Andrew J. Tatem, Natalia Tejedor-Garavito.

**Methodology:** Adelle Wigley, Josh Lorin, Dan Hogan, C. Edson Utazi, Andrew J. Tatem, Natalia Tejedor-Garavito.

**Project administration:** Natalia Tejedor-Garavito.

**Supervision:** Andrew J. Tatem.

**Validation:** Josh Lorin, Dan Hogan.

**Visualization:** Adelle Wigley, Natalia Tejedor-Garavito.

**Writing – original draft:** Adelle Wigley.

**Writing – review & editing:** Adelle Wigley, Josh Lorin, Dan Hogan, C. Edson Utazi, Brittany Hagedorn, Emily Dansereau, Andrew J. Tatem, Natalia Tejedor-Garavito.

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
