## [Decision Letter · Decision Letter 0]

17 Aug 2022

PGPH-D-22-01112

Estimates of the number and distribution of zero-dose and under-immunised children across remote-rural, urban, and conflict-affected settings in low and middle-income countries

Dear Dr. Wigley,

Thank you for submitting your manuscript to PLOS Global Public Health. After careful consideration, we feel that it has merit but does not fully meet PLOS Global Public Health’s publication criteria as it currently stands. Therefore, we invite you to submit a revised version of the manuscript that addresses the points raised during the review process.

The reviewers have very minimal feedback. Please respond to the minor issues below. Note that one reviewer mentioned your abstract. Although PLOS Global Public Health does not use structured abstract headings, it might be worth adding a bit more methods into the abstract in your response to that reviewer's comment.

We look forward to receiving your revised manuscript.

Kind regards,

Abram L. Wagner, PhD, MPH

Academic Editor

Journal Requirements:

1. Please update the completed 'Competing Interests' statement. Please declare all competing interests beginning with the statement “I have read the journal's policy and the authors of this manuscript have the following competing interests:”.

2. Please amend your online detailed Financial Disclosure statement. This is published with the article. It must therefore be completed in full sentences and contain the exact wording you wish to be published.

3. Please provide separate figure files in .tif or .eps format only and ensure that all files are under our size limit of 10MB.

4. We have noticed that you have a list of Supporting Information legends in your manuscript. However, there are no corresponding files uploaded to the submission. Please upload them as separate files with the item type 'Supporting Information'.

5. All figures and supporting information files will be published under the Creative Commons Attribution License (creativecommons.org/licenses/by/4.0/). Authors retain ownership of the copyright for their article and are responsible for third-party content used in the article. 

Figures 1, 2, 3, and 5: please (a) provide a direct link to the base layer of the map used and ensure this is also included in the figure legend; (b) provide a link to the terms of use / license information for the base layer. We cannot publish proprietary or copyrighted maps (e.g. Google Maps, Mapquest) and the terms of use for your map base layer must be compatible with our CC-BY 4.0 license. 

Please upload any written confirmation as an 'Other' file type. It must clarify that the copyright holder understands and agrees to the terms of the CC BY 4.0 license; general permission forms that do not specify permission to publish under the CC BY 4.0 will not be accepted. Note that uploading an email confirmation is acceptable.

Additional Editor Comments (if provided):

Reviewers' comments:

Reviewer's Responses to Questions

**Comments to the Author**

1. Does this manuscript meet PLOS Global Public Health’s publication criteria? Is the manuscript technically sound, and do the data support the conclusions? The manuscript must describe methodologically and ethically rigorous research with conclusions that are appropriately drawn based on the data presented.

Reviewer #1: Yes

Reviewer #2: Yes

Reviewer #3: Partly

Reviewer #4: Partly

2. Has the statistical analysis been performed appropriately and rigorously?

Reviewer #1: I don't know

Reviewer #2: Yes

Reviewer #3: Yes

Reviewer #4: No

3. Have the authors made all data underlying the findings in their manuscript fully available (please refer to the Data Availability Statement at the start of the manuscript PDF file)?

Reviewer #1: Yes

Reviewer #2: Yes

Reviewer #3: Yes

Reviewer #4: No

4. Is the manuscript presented in an intelligible fashion and written in standard English?

Reviewer #1: Yes

Reviewer #2: Yes

Reviewer #3: Yes

Reviewer #4: Yes

5. Review Comments to the Author

Reviewer #1: Summary:

This study uses geospatial datasets to map and estimate global and regional gaps in under one vaccination status with a focus on settings expected to be high risk for un- or under-vaccinated children: remote-rural, urban, and conflict affected areas. This study found that although many of the un- and under-vaccinated are in these high risk areas, about 60% live outside of those settings not classified as remote (rural non-remote), pointing to the need for more targeted strategies at the national and regional level to close vaccination gaps.

Review:

This is a high caliber, well-written and impactful study. This study’s approach is both novel and practical, using multiple geospatial datasets to drill down to assess vaccination status in settings hypothesized to be high risk. The methods are sound and clearly written, including the acknowledgment of limitations in slum estimates. The results are well-presented with easy to follow figures, providing key estimates and illustrating vaccination gaps overall, remote-rural, urban, conflict areas, and rural non-remote. The discussion interprets the results well, with a clear and concise summary of the key findings: although many of the vaccination gaps exist in the expected high-risk settings, there is great variability in vaccination coverage within countries. The discussion also provides important recommendations and strategies to improve vaccination coverage. Overall, this study adds scientific value with detailed estimates of vaccination gaps based on geospatial data, uncovers gaps in unexpected settings, and shows how this geospatial data approach can be used to improve vaccination coverage in targeted settings.

Reviewer #2: Thank you, authors, for the opportunity to review your work. The work is very interesting and adds detailed input to existing knowledge. I just have a few concerns highlighted below;

Line 25 …….estimate…. replace with ….estimated….

Line 28 ….. we find… replace with ………we found…..

Line 29 delete We estimate…. It should read as Of the…….

Lines 57-61 is just one sentence consider breaking it into 2 or 3 sentences

Overall, the article was well written. I would just advise the authors to revise some of the long sentences and make them shorter as it is easier to engage with shorter sentences than long sentences. Well done.

Reviewer #3: Abstract

Redesign the entire abstract following the IMRaD structure. Start from the general to the specific in the introduction. The methodology should also be included in the summary. Clearly state your results (rephrasing). The conclusion must be well separated and must support the results of the work.

Results

Put the titles of the tables at the top

Remove the percentage symbol inside table 1 by adopting for example n(%). This form of presentation is easy to understand. In the header of table 1, put for example DTP1 (N=14,030,486) and so on for the other variables in the header.

Review the references. I noticed that the numbers are missing. Also review the pages.

For example, on line 497 for reference 33, it is ............26(12):1-4.

Reviewer #4: Title:

Remove one 'and' and just use comma. Hence the title should be: 'Estimates of the number, distribution of zero-dose and under-immunised children

across remote-rural, urban, and conflict-affected settings in low and middle-income

countries

Methods and results:

only Univariate level i.e,. propotions have been calculated. Can you go farther to the level of bivariate and multivariate leves ie. to find out Asociations of the factors?

6. PLOS authors have the option to publish the peer review history of their article (what does this mean?). If published, this will include your full peer review and any attached files.

**Do you want your identity to be public for this peer review?** For information about this choice, including consent withdrawal, please see our Privacy Policy.

Reviewer #1: No

Reviewer #2: No

Reviewer #3: **Yes: **Nicolas Hamondji AMEGAN, MPH

Reviewer #4: **Yes: **Dr. Joseph C. Hokororo

---

## [Editor Report · Decision Letter 1]

6 Oct 2022

Estimates of the number and distribution of zero-dose and under-immunised children across remote-rural, urban, and conflict-affected settings in low and middle-income countries

PGPH-D-22-01112R1

Dear Ms Wigley,

We are pleased to inform you that your manuscript 'Estimates of the number and distribution of zero-dose and under-immunised children across remote-rural, urban, and conflict-affected settings in low and middle-income countries' has been provisionally accepted for publication in PLOS Global Public Health.

Best regards,

Abram L. Wagner, PhD, MPH

Academic Editor